# An Integrative Study of Aortic mRNA/miRNA Longitudinal Changes in Long-Term LVAD Support

**DOI:** 10.3390/ijms22147414

**Published:** 2021-07-10

**Authors:** Dana Dlouha, Peter Ivak, Ivan Netuka, Sarka Benesova, Zuzana Tucanova, Jaroslav A. Hubacek

**Affiliations:** 1Experimental Medicine Centre, Institute for Clinical and Experimental Medicine, 140 21 Prague, Czech Republic; jahb@ikem.cz; 2Department of Cardiovascular Surgery, Institute for Clinical and Experimental Medicine, 140 21 Prague, Czech Republic; ivap@ikem.cz (P.I.); ivne@ikem.cz (I.N.); dolz@ikem.cz (Z.T.); 3Department of Physiology, 3rd Faculty of Medicine, Charles University, 100 00 Prague, Czech Republic; 4Second Department of Surgery, Department of Cardiovascular Surgery, 1st Faculty of Medicine, Charles University, 121 08 Prague, Czech Republic; 5Laboratory of Informatics and Chemistry, Faculty of Chemical Technology, University of Chemistry and Technology, 166 28 Prague, Czech Republic; sarka.benesova@ibt.cas.cz; 6Laboratory of Gene Expression, Institute of Biotechnology CAS, BIOCEV, 252 50 Vestec, Czech Republic; 73rd Department of Internal Medicine, 1st Faculty of Medicine, Charles University, 121 08 Prague, Czech Republic

**Keywords:** mRNA, microRNA, aorta, mechanical circulatory support, left ventricular assist device

## Abstract

Studying the long-term impact of continuous-flow left ventricular assist device (CF-LVAD) offers an opportunity for a complex understanding of the pathophysiology of vascular changes in aortic tissue in response to a nonphysiological blood flow pattern. Our study aimed to analyze aortic mRNA/miRNA expression changes in response to long-term LVAD support. Paired aortic samples obtained at the time of LVAD implantation and at the time of heart transplantation were examined for mRNA/miRNA profiling. The number of differentially expressed genes (Pcorr < 0.05) shared between samples before and after LVAD support was 277. The whole miRNome profile revealed 69 differentially expressed miRNAs (Pcorr < 0.05). Gene ontology (GO) analysis identified that LVAD predominantly influenced genes involved in the extracellular matrix and collagen fibril organization. Integrated mRNA/miRNA analysis revealed that potential targets of miRNAs dysregulated in explanted samples are mainly involved in GO biological process terms related to dendritic spine organization, neuron projection organization, and cell junction assembly and organization. We found differentially expressed genes participating in vascular tissue engineering as a consequence of LVAD duration. Changes in aortic miRNA levels demonstrated an effect on molecular processes involved in angiogenesis.

## 1. Introduction

The use of continuous-flow left ventricular assist devices (CF-LVADs) in patients with end-stage heart failure has become a widely used and sustainable treatment strategy, both as a bridge to transplant (BTT) and as destination therapy (DT). CF-LVAD may induce pathological changes to the aortic wall and aortic valve [1]. Important histologic changes in the aortic wall, before and after CF-LVAD implantation, with degeneration of smooth muscle cells and elastic fibers, were previously reported [2]. CF-LVADs may contribute to the deterioration of aortic functional parameters (e.g., aortic stiffness) or structural changes (e.g., increase of wall thickness or collagen content) through adverse effects of the nonphysiological flow [3]. The nonpulsatile flow of CF-LVAD generates dynamic remodeling within the aorta. Remodeling within the aortic root and proximal ascending aorta may also contribute to the pathophysiology of aortic regurgitation with CF-LVAD. Studies have demonstrated that the proximal aorta dilates after chronic CF-LVAD support, and that an increasing aortic diameter was associated with the development of aortic regurgitation [4].

MiRNAs are endogenous small noncoding RNAs that regulate mRNA translation of target genes through the RNA interference pathway, strongly influencing a wide range of cellular processes and biological pathways [5]. As such, miRNAs are fine-tuners of gene expression patterns in response to pathophysiological stimuli. Most miRNAs are more ubiquitously expressed and are not cell-type specific. Thus, many miRNAs are expressed at relatively low levels under basal conditions, but during pathological stress, are strongly upregulated [6]. Changes in tissue miRNA expression levels due to blood flow can potentially affect networks of genes regulating endothelial and vascular smooth muscle cell (SMC) function, inflammation, and atherosclerosis [7].

Alteration in gene expression reflects changes in cellular function and behavior, in development, and disease states. The major cardiovascular diseases, including coronary artery disease, myocardial infarction, congestive heart failure and common congenital heart disease, are caused by multiple genetic and environmental factors, as well as the interactions between them. The underlying molecular pathogenic mechanisms for these disorders are still largely unknown, but gene expression may play a central role in the development and progression of cardiovascular disease [8]. Gene expression analysis can also contribute to understanding and discovering novel and sensitive biomarkers of cardiovascular disease. Over the past two decades, methods of measuring gene expression have improved dramatically with a plethora of hybridization arrays available, followed by RNA-Seq, the sequencing of short or long RNA reads using massively parallel sequencing technology [9].

Studying the long-term use of LVADs offers an opportunity for a complex understanding of the pathophysiology of vascular changes in patients with mechanical circulatory support, which produce nonphysiological blood flow patterns. The aim of the present study was therefore to detect accurate mRNA/miRNA associations in the aorta in response to long-term LVAD. 

## 2. Results

The principal component analysis of expression profiles segregated samples before and after LVAD support (Appendix A). The mRNAs profile (Figure 1A) demonstrated differentially expressed genes (DEGs). We identified a total of 277 DEGs (Pcorr < 0.05) after long-term LVAD support. 141 DEGs were upregulated in aortic tissue after LVAD support. Between twenty DEGs, we identified Collagen Type I Alpha 2 Chain (*COL1A2*); Chemokine-like receptor 1 **(***CMKLR1*); S100 calcium-binding protein A4 (*S100A4*); Elastin (*ELN*); and Collagen Type III Alpha 1 Chain (*COL3A1*) to be upregulated (all Pcorr < 0.0001). Solute Carrier Family 12 Member 2 (*SLC12A2*); Pyruvate Dehydrogenase Kinase 4 (*PDK4*); Inhibitor of DNA Binding 1 (*ID1*); Arginine and Glutamate Rich 1 (*ARGLU1*) and noncoding RNA metastasis-associated lung adenocarcinoma transcript 1 (*MALAT1*) were downregulated (all Pcorr < 0.0001, Appendix A). The Wald test performed on samples after support to assess the effect of HeartMate II (HM II), and HeartMate 3 (HM 3) devices did not show any significantly differentially expressed genes (Appendix A).

Gene ontology (GO) analysis identified 31 biological processes in GO terms enriched in explanted tissue (false discovery rate (FDR) q-value < 0.001), predominantly involved in extracellular matrix organization and collagen fibril organization. Moreover, 17 molecular functions of GO terms, and 10 cellular components of GO terms were also enriched (Appendix A).

The whole miRNome profile revealed 69 DE miRNAs (Pcorr<0.05, after Bonferroni correction (BFC), Figure 1B). A total of 30 miRNAs were upregulated (Appendix A), and the most increased expression was detected for let-7a/d/e/f; miR-181a/b; miR-29b; miR-149 and miR-99b (all Pcorr < 0.01). The opposite effect of LVAD was found on miR-19a/b; miR-654; miR-664a; miR-885; and -511 (all Pcorr < 0.01).

With the usage of miRWalk and miRDB, 170 DEGs were identified as potential targets of upregulated miRNAs and 159 DEGs as potential targets of downregulated miRNAs. From these, 138 DEGs were possible targets of both group miRNAs (Appendix A). 

In order to discover the relationship between deregulated miRNAs and mRNAs, Pearson correlation between differentially expressed miRNAs and all potential target mRNAs was assessed (Figure 2). This integrated mRNA/miRNA analysis revealed that potential targets of miRNAs upregulated in explanted samples are mainly involved in GO biological process terms related to cell junction assembly and cell junction organization (FDR q-value < 0.05, Appendix A). Enriched GO terms in targets of downregulated miRNAs were related to dendritic spine organization and neuron projection organization (FDR q-value < 0.05, Appendix A). Moreover, an enriched Kyoto Encyclopedia of Genes and Genomes (KEGG) pathway related to endocytosis was found (FDR q-value < 0.05, Appendix A).

## 3. Discussion

Our multiple regression analysis is, to our best knowledge, the first study focused on changes in mRNA/miRNA expression in paired aortic samples collected before LVAD implantation and at the time of LVAD explantation, during heart transplantation (HTx).

We compared gene expression profiles using a robust and simple mRNA sequencing method. We found the significant DEGs were predominantly involved in extracellular matrix (ECM) and collagen fibril organization in aortic tissue after LVAD explantation. ECM is an active and dynamic structure with a fundamental role in regulating vascular function in normal and pathological conditions. Homeostasis of the vascular ECM may affect intrinsic properties of the arterial wall and arterial stiffness [10]. The ECM is a key component of the local cellular microenvironment. It comprises structural proteins (e.g., elastin and collagen), proteoglycans, and glycosaminoglycans. Among the different cell types, smooth muscle cells (SMCs) and fibroblasts are examples of cells that produce significant ECM. Interestingly, Coffey et al. [11], using analysis of the integrated miRNA/mRNA network, identified pathways predominantly involved in extracellular matrix function in patients affected by aortic stenosis.

The previously reported histologic analysis showed significant degenerative changes in the aortic wall, SMCs disorientation and depletion, elastic fiber fragmentation and depletion, medial fibrosis, and atherosclerosis changes in ascending aortic tissue at the follow-up than at device implantation [2]. Furthermore, there was evidence of structural remodeling within the aortas of CF-LVAD patients, including an increase in total wall thickness, an increase in collagen content, and reduced elastin content that may explain the increase in vessel stiffness [12]. We identified elevated expression of collagens *COL1A1* and *COL3A1*. Collagen and elastin are the most abundant ECM proteins of the aortic wall, and they are responsible for characteristic mechanical properties—tensile strength and elasticity. Over-accumulated collagen in the aorta may lead to medial fibrosis, hypothetically resulting in decreased arterial distensibility [13]. Surprisingly, we also detected elevated expression of *ELN*, which is usually lowered in processes related to atherosclerosis, such as pathological flow. We may only hypothesize whether this phenomenon can be attributed to compensatory mechanisms reacting on the non-physiological flow pattern or should be a subject for further research, as the reason for this observation is unclear at this stage of research.

We found overexpressed *S100A4*, a member of the large family of S100 proteins, under LVAD support. S100A4 controls different cellular pathways, exerting numerous effects on processes that are cell- and tissue-type dependent. In activated fibroblasts, endothelial, dendritic, and mast cells, as well as in macrophages, monocytes, neutrophils, and T-lymphocytes, S100A4 has a significant role in stimulating invasion and migration, cytoskeletal dynamics and in promoting proinflammatory phenotypes [14]. S100A4 represents a well-known marker that characterizes a complex biological process where endothelial cells assume a mesenchymal phenotype, known as the “endothelial-to mesenchymal transition”, changing morphology and functions, acquiring accentuated motility and contractile properties, typical of fibrotic processes [15]. Furthermore, upregulated *CMKLR1* is currently the only chemerin receptor. CMKLR1 receptor, and the proposed pro- and anti-inflammatory properties of chemerin, suggest a role of this adipokine in inflammatory states and possibly atherosclerosis. It was reported that foam cell *CMKLR1* expression strongly and positively correlates with aortic atherosclerosis, but only marginally with coronary atherosclerosis [16]. Our findings of aortic increased expression levels of *COL1A2*; *CMKLR1*; *S100A4*; and *COL3A1* may support previously mentioned morphological changes in the aorta under CF-LVAD.

Changes in aortic wall functional properties as the possible consequence of a pulsatility decrement caused by implantation of CF-LVAD were described [3]. Patel et al. reported that patients with CF-LVADs before heart transplant had an increase in proximal aortic stiffness compared with patients without an LVAD or with pulsatile flow LVADs before transplant [3,17]. DEGs involved in ECM organization could also suggest a potential link with the development of acquired aortic insufficiency (AI), a significant complication that develops following the implantation of CF-LVAD [18,19]. We detected downregulated *MALAT1*, a gene coding stiffness-sensitive long non-coding RNA. This non-coding RNA regulates stiffness-dependent VSMC proliferation and migration [20], which may influence aortic functional properties.

Despite the advantages and improving results of the CF-LVAD therapy, the loss of pulsatility may lead to different complications on the micro and macrovascular levels. Vascular changes may be linked with the occurrence of clinically adverse events related to CF-LVAD therapy, such as non-surgical bleeding, e.g., gastrointestinal bleeding related to arteriovenous malformations [21] or von Willebrand factor (vWF) deficiency [22], or other clinical complications such as cerebrovascular events [23,24], device thrombosis [25,26] or development of aortic insufficiency [27,28].

One of the pathogenetic mechanisms of cardiovascular complications with CF-LVADs may be endothelial dysfunction. Endothelial dysfunction is related to heart failure in general. After the implantation of the device, the endothelial dysfunction does not improve and may even deteriorate [29]. In our miRNAs profile, we identified deregulated multiple miRNAs involved in vascular remodeling (Table 1) which potentially involved endothelial dysfunction progression. Several miRNAs have been shown to control the varying mechanisms which govern SMC plasticity [30]. In response to LVAD, we found upregulated miRNAs that influence SMC dynamics and downregulated miRNAs known to stimulate apoptosis during atherosclerosis plaque development. Leeper et al. reported that chronic SMC apoptosis accelerates vascular disease progression, promotes calcification, and induces features of medial degeneration, like atrophy, elastin fragmentation, and enhanced glycosaminoglycan deposition, thus worsening endothelial dysfunction [30]. 

Several authors, including Morgan et al., observed worsened endothelial function in long-term CF-LVAD patients [31]. In our study, we observed dysregulation in miRNAs participating in the regulation of vascular development, growth, and differentiation [32], which may indicate the role of these miRNAs in the further development of endothelial dysfunction. Interestingly, we also identified upregulated miRNAs that regulate mRNAs encoded by genes in human endothelial cells related to vascular function and blood pressure regulation [33].

Correlation analysis revealed that miR-409-3p (upregulated in explanted tissue), one of the most potent fibrinogen downregulating miRs [45], potentially affects the expression of the highest number of genes (10 genes with a correlation coefficient higher than |7|). However, only two genes correlated negatively, suggesting they may be direct targets (*ZEB1* and *RAP1A*). *ZEB1* gene is associated with the regulation of vasculogenesis [46], whereas *RAP1A* promotes angiogenesis and dynamic regulation of endothelial barrier [47].

It should be mentioned that in our study, patients with two types of LVAD were studied—an axial-flow LVAD HeartMate II and the HeartMate 3, a centrifugal-flow pump with intrinsic artificial pulsatility. Nevertheless, this intrinsic pulsatility was originally designed to enhance pump washout and prevent blood stasis and thrombosis. Our previous work found that this pulsatility does not avert endothelial dysfunction [29,48]. Therefore, the HeartMate 3 should also be considered as CF-LVAD. We are aware that our population of patients is not exceedingly large; nevertheless, we did not observe any differences between the pump types.

The vasculature is one of the most dynamic tissues that encounter numerous mechanical cues derived from pulsatile blood flow, blood pressure, the activity of smooth muscle cells in the vessel wall, and the transmigration of immune cells [49]. Endothelial cell junction assembly and cell junction organization play pivotal roles in tissue integrity, barrier function, and cell–cell communication, respectively [50]. In this study, a multistep approach combining mRNA and miRNA expression profiles and bioinformatics analysis was adopted to identify the mRNA/miRNA regulatory network. Enriched GO terms in targets of upregulated miRNAs were related to cell junction assembly and cell junction organization.

## 4. Materials and Methods

### 4.1. Subjects

All examined individuals provided their informed consent, which the institution’s ethics committee approved together with the study protocol. The protocol of this study was conducted according to the principles of the Declaration of Helsinki [51].

A total of 16 patients (median age 57 years, range 18–65) who required mechanical circulatory support from HeartMate II (*N* = 4) and HeartMate 3 (*N* = 12) as a bridge to transplantation or destination therapy from July 2015 to March 2018 at the Institute for Clinical and Experimental Medicine in Prague, were enrolled in our study. The etiology of heart failure was predominantly nonischemic dilated cardiomyopathy (*N* = 13). The median LVAD support duration was 382 days (ranging from 162 to 887 days). The basic characteristics of the patients are summarized in Table 2.

### 4.2. Sampling

Paired aortic tissue was obtained at the time of LVAD implantation and at the time of HTx from CF-LVAD patients. Approximately 30 mg of tissue was immediately after exclusion from aorta inserted into RNase/DNase-free tubes pre-filled with All Protect Tissue Reagent (Qiagen GmBH Strasse 1, Hilden, Germany). Samples were stored for 2–4 weeks at 4 °C and then at −80 °C before RNA extraction. The storage time of tissue ranged from 33 to 876 days.

### 4.3. mRNA and miRNA Analysis

Total RNA, including miRNA, was extracted from 10 mg of aortic tissue according to protocol using the miRCURY^TM^ RNA isolation kit for tissue (Qiagen GmBH Strasse 1, Hilden, Germany). RNA quality and quantity were assessed using a Fragment Analyzer system (Agilent technologies, 301 Stevens Creek Blvd., Santa Clara, CA, USA).

Gene expression was measured in paired samples from 10 patients. QuantSeq 3′ mRNA sequencing for RNA quantification was performed using a high-throughput technique using 3′mRNA-Seq Library Prep Kit FWD and 3′mRNA-Seq Library Prep Kit REV (https://www.lexogen.com/quantseq-3mrna-sequencing/; accessed on 20 March 2020) at Lexogen (Campus Vienna Biocenter 5, 1030 Vienna, Austria). QuantSeq. 3′ mRNA library preparation predominantly produces fragments for sequencing close to the 3′ end of polyadenylated mRNA, generally from the last exon and the 3′ untranslated region (3 UTR) [52]. The total RNA input was 20 ng. There was no prior poly(A) enrichment or rRNA depletion. The QuantSeq Forward kit has an oligo (dT) primer containing the Illumina-specific Read 2 linker (P7), which is annealed to the 3′ end of the mRNA fragment to synthesize the first cDNA strand via reverse transcriptase. The second strand synthesis is commenced by random priming and DNA polymerase extension. The random primer contains the Illumina-specific Read 1 linker sequence (P5). Sequencing commences from the Read 1 sequencing primer and goes toward the poly(A) tail with only one fragment produced per transcript [9].

SYBR green-based real-time quantitative PCR (RT-PCR) for miRNA profiling (in a total of 16 patients) was performed using miRNome Panels (Qiagen GmBH Strasse 1, Hilden, Germany). Passive Reference Dye (ROX^TM^ 30 nm) was included for all PCR reactions. Measurement was performed using the QuantStudio6 Flex instrument (ThermoFisher Scientific, 81 Wyman Street, Waltham, MA, USA). Inter-plate calibrators (IPC) for calibration between PCR plate runs, and spike-in controls to ensure the quality of RNA isolation, cDNA synthesis reaction, and PCR was included in each measurement [53].

### 4.4. Processing of mRNA Sequencing Data

QuantSeq 3′ mRNA sequencing produced 140 million reads with 35–76 bp length. Raw reads were trimmed of bases with Phred 33 quality lower than 30 using Cutadapt software v2.9 [54]. Reads mapping to rRNA and UniVec (common vector contaminations in RNA sequencing) databases were discarded. Mapping was performed using a bowtie aligner [55], with one mismatch allowed. The remaining reads were mapped with STAR aligner to the human genome (GRCh 38.95), with one mismatch allowed [56]. Only uniquely mapping reads were assigned to individual genes using featureCounts software [57].

### 4.5. DESeq2 Analysis of mRNA Expression

Genes with less than 10 counts per all samples were removed before further analysis. Raw counts were normalized using the median ratio method built in DESeq2 software [58]. To account for paired samples, parameters included in the DESeq2 model were patient IDs and conditions before (implant) and after (explant) LVAD. Differential expression between implant and explant samples was tested with one parameter Wald test built in DESeq2; *p* values were adjusted for multiple testing with Bonferroni correction (BFC). A *p*-value < 0.05 was considered statistically significant.

### 4.6. Gene Ontology Analysis

Gene ontology was performed using Gene Ontology enRIchment anaLysis and visuaLizAtion tool (Gorilla), a web-based tool [59]. All genes expressed in measured samples (threshold > 10 raw counts per all samples) were ranked using the following formula –log (*p*-value) *** log2FC; the resulting list was used as input for calculating the *p*-value of the minimum hypergeometric score, as described in detail by [60]. KEGG pathway analysis was performed with clusterProfiler tool using 10,000 permutations and a gene set size between 3–800 genes. As input for KEGG pathway analysis, the same ranked list of genes was used for GO analysis [61,62].

### 4.7. miRNA Profile Analysis

Gene Expression software (GenEx SW, Multid Analysis AB, Göteborg, Sweden) was used for miRNA expression analysis. Ct values higher than 35 were replaced by 35. A total of 330 miRNAs with a call rate <40% (i.e., more than 60% data are invalid of that miRNA) were removed from further analysis. The missing data, exceedingly low miRNA levels, were replaced by deltaCt + 2 (representing at least 1/4 of the detectable miRNAs amount). Data were normalized with the mean expression of all miRNAs and converted to relative quantities and Log2. *P* values were corrected for multiple testing with BFC. A *p*-value < 0.05 was considered statistically significant.

### 4.8. Integrated mRNA/miRNA Analysis

Differentially expressed miRNAs were used to predict mRNA targets in miRWalk (http://mirwalk.umm.uni-heidelberg.de/search_mirnas/; accessed on 18 October 2020) and miRDB (http://www.mirdb.org/; accessed on 18 October 2020) databases. miRDB target prediction was restricted to gene targets with prediction scores less than 60, and miRNAs with more than 2000 genes in the genome were excluded. For prediction in the miRWalk database, no restrictions were used. All genes expressed in our samples, which appeared in the predicted targets in either of the databases, were used for correlation analysis. Pearson correlation was used to correlate log_10_ transformed normalized expression values of differentially expressed miRNAs with log_10_ transformed normalized expression values of expressed genes. Only pairs with a correlation better than −0.7/0.7 were considered for further GO analysis and KEGG pathway. GO analysis and KEGG pathway were performed using clusterProfiler with the same parameters described in Section 4.6.

## 5. Conclusions

The study provides additional insight into the pathophysiology of vascular changes observed in patients after LVAD implantation. Significant regulation of mRNAs involved in ECM and collagen fiber organization in response to the implantation of LVAD was observed, which may suggest infliction of ECM homeostasis resulting in changes of intrinsic properties of the vascular wall and arterial stiffness.

## Figures and Tables

**Figure 1 ijms-22-07414-f001:**
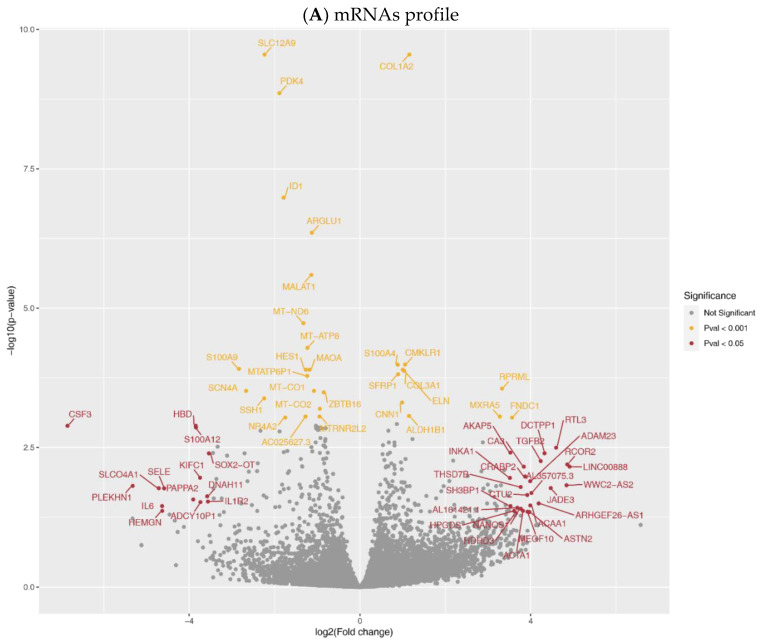
Expression profiles in aortic tissue. Volcano plot of comparative mRNA resp. miRNAs expression profiles in samples before and after LVAD support; x-axis indicates difference in expression level on a log2 scale; y-axis represents corresponding *P*-values on a negative log scale. (**A**) mRNA profile: yellow points indicate mRNAs with Pcorr < 0.001; red points indicate mRNAs with Pcorr < 0.05, & Pcorr > 0.001, & |logfc| > 3.5.; (**B**) miRNA profile: yellow points indicate miRNAs with Pcorr < 0.001; red points indicate miRNAs with Pcorr < 0.05, & Pcorr > 0.001.

**Figure 2 ijms-22-07414-f002:**
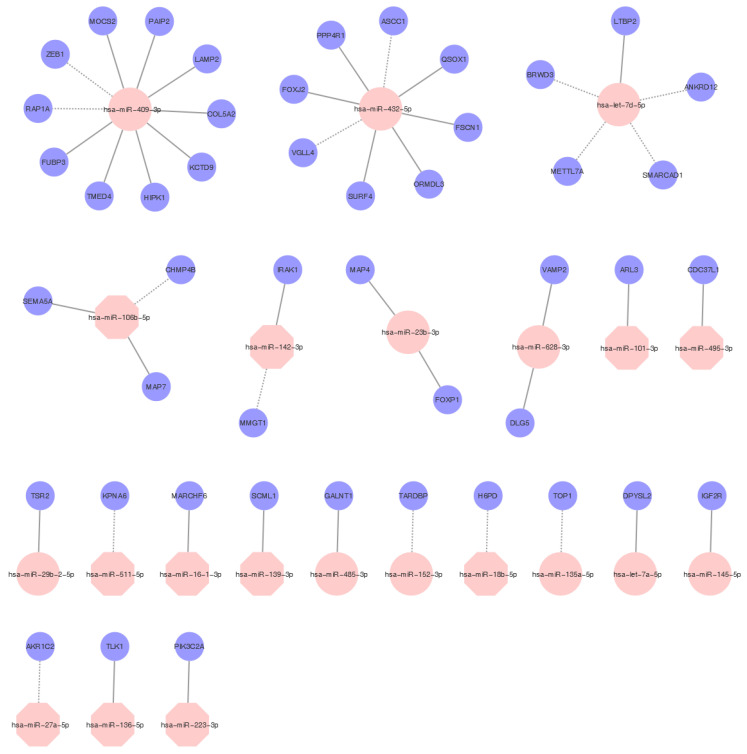
MiRNAs regulatory network. Examples of flexible and selective regulatory networks between miRNAs and mRNAs in samples after LVAD support. Selected overexpressed miRNAs are highlighted in pink circles, and underexpressed miRNAs are highlighted in pink octagons. The targets are highlighted in blue circles. The full line indicates a positive correlation, and the discontinued line indicates a negative correlation with target mRNAs.

**Table 1 ijms-22-07414-t001:** miRNAs participated in vascular remodeling in aortic tissue.

miRNA	Expression	Target (s)	Function
miR-23b	**up**	*TLP3*, *FOXO4*, *CHI3L1*, *SMAD3*	SMCs proliferation, differentiation, cytokine production
miR-29b	**up**	*COL1A1*, *COL3A1*, *COL5A1*, *ELN*, *MMP2*, *MMP9*, *PTEN*, *ADAMTS7*	ECM production, SMCs proliferation, arterial calcification, cell apoptosis
miR-155	**up**	*SMAD*, *BCL6*, *CTLA4*, *MMP1*, *MMP3*, *SOCS*, *NF-κB signaling transcription factor*	SMCs differentiation, regulation of inflammation
miR-206	**up**	*ARF6*, *SLC8A1*	SMCs differentiation
miR-34a	**down**	*SIRT1*, *NOTCH*	SMCs proliferation, differentiation
miR-145	**up**	*KLF4/5*, *MYOCD*, *ELK1*, *SRF*, *SOX9*	SMCs differentiation, proliferation
			Inhibits TGF-β signaling, ECM production, regulation of fibrosis
miR-19a/b	**down**	*FZD4*, *LRP6*, *TLR2*, *TGFBRI/TGFBRII*	ECs proliferation, differentiation, angiogenesis, WNT signaling pathway, regulation of fibrosis
miR-20a	**down**	*MKK3*, *TLR4*	Reduction of ECs migration and angiogenesis, TXNIP signaling, inflammation
miR-149	**up**	*FGFR1*, *GPC1*	Regulation of angiogenic functions of ECs
Let-7a/c/e/f	**up**	*TGFBR3*, *TBX5*, *ADRB1*, *EDN1*, *FGF5*, *IL6*, *IκBβ*	Regulation of angiogenesis of ECs and inflammation
miR-100	**up**	*mTOR*, *NOX4*	Regulation of neovascularization
miR-99b	**up**	*NOX4*, *TGFβ*	Differentiation of ECs
miR-30c/e	**up**	*CTGF*	Promotion of the synthesis of ECM and collagen, regulation of fibrosis
miR-142-3p	**down**	*ADAM9*, *HMGB1*, *AZIN1*, *JNK1*	Regulation of fibrosis
miR-15b/16	**down**	*TGF-βR1*, *p38*, *SMAD3*, *SMAD7*, *ENDOGLIN*, *AKT3*	Regulation of fibrosis, cell apoptosis, and angiogenesis
miR-885	**down**	*ULK2*	Cell autophagic processes
miR-511	**down**	*FOXC1*	Regulation of angiogenesis
miR-664a	**down**	*TGFBR2*, *AKT*	Inhibits TGF-β signaling, ECM production, regulation of fibrosis
miR-654	**down**	*PTEN*, *ATM*, *ADAM10*, *RAB22A*	Regulation of fibrosis and inflammation

For more details see [30,31,32,33,34,35,36,37,38,39,40,41,42,43,44].

**Table 2 ijms-22-07414-t002:** Demographics of the patients selected for mRNA/miRNA analysis.

N (female %)	16 (18.8%)
Age (years)	49.6 ± 16.8
BMI (kg/m^2^)	25.6 ± 5.5
Diabetes mellitus (%)	1 (6.3%)
Hypertension (%)	8 (50%)
Hyperlipidemia (%)	5 (31.3%)
CVA/TIA (%)	0
NYHA classification IV (%)	11 (68.8%)
Etiology of nonischemic DCM (N)	13
Idiopathic	11
Familial	1
Toxic	1
Etiology of hypertrophic cardiomyopathy	1
Etiology of ischemic DCM (N)	1
Etiology of noncompact DCM (N)	1
CRTD/ICD before LVAD implant, %	12 (75%)
Type of LVAD	
Heart Mate II	4
Heart Mate 3	12
Days of LVAD support (days)	382 (325.5)

HF, heart failure; CVA, cerebrovascular accident; TIA, transient ischemic attack; NYHA, New York Heart Association; DCM, dilated cardiomyopathy; CRTD, cardiac resynchronization therapy defibrillator; ICD, implantable cardioverter-defibrillator. Categorical data are presented as number (%), continuous data as mean ± SD or median (IQR), respectively. Familial DCM is only defined if the patient has one or more family members diagnosed with idiopathic DCM or has a first-degree relative who experienced sudden unexplained death under 35 years. Days of LVAD support are based on patients who already underwent HTx.

## Data Availability

All data that support the findings of this study are available from the corresponding author upon reasonable request.

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
