# Peer review of "An Integrative Study of Aortic mRNA/miRNA Longitudinal Changes in Long-Term LVAD Support"

_ijms, 2021, doi:10.3390/ijms22147414_

Round 1
Reviewer 1 Report
The paper cover important topic of the detection of the accurate mRNA/miRNA associations in the aorta in response to long-term LVAD.
-
- Introduction. Please add the information about gene expression application for the heart diseases investigation
- Materials and Methods should be moved after Introduction
- Discussion. As we know the HM 3 can generate pulsations in contrast with HM II, could you discuss about how such changes in the flow behavior react on mRNA/miRNA profiling
Reviewer 2 Report
Dlouha and colleagues prepared an interesting study that examines the association between duration of LVAD use and changes in mRNA/miRNA expression in aortic tissue on a patient-level analysis. Although the study purpose is well founded, the included patients do not fit the study criteria. It is framed as a study that evaluates the long-term impact of continuous-flow LVAD, however 75% of the patients included (12/16) had a Heartmate 3 LVAD, which is a categorically pulsatile-flow system and not a continuous-flow system. If the authors intend to evaluate the effects of non-physiologic continuous-flow LVAD on aortic mRNA/miRNA expression, then inclusion of HM3 patients is antithetical to this goal.
Reviewer 3 Report
The authors investigated the gene expression ( mRNA/miR) profile in paired aortic biopsy from patients with heart insufficiency before implantation of a LVAD device and at time of its removal after 162 to 382 days. They aimed to detect possible changes in miR/mRNA expression due to long term LVAD support. They found in total 277 differences in mRNA and 69 in miR expression involved in cell structural stability, communication, angiogenesis. Indeed this seems to be the first study on this subject using patients aorta showing responses due to a long term LVAD support. This study is really interesting, though effects on aortic structural integrity could be expected. It is however, only a first step, the important question to be answered should be: are these observed changes or some changes reversible with time after explantation of the device? What are the consequences of these findings for the patients?Unfortunately discussion of these important points are missing. There are further several major and minor concerns to be considered prior to publication.
Major concerns
1) Introduction, second paragraph, lines 50ff: The descriptions on miRs are misleading. mi Rare very important regulators of protein synthesis also under normal conditions not only in cardiovascular diseases. And please when copying sentences from the literature cited, please copy it correctly! Line 53 : Most miRs are more ubiquitously expressed and are not cell-type specific. Is the correct sentence in Thums manuscript. Also, the next sentence is wrong. At first it is not an explanation of the preceeding statement, second the authors mix up the meaning of regulation and upregulation. Certainly the actions/expression of miRs have to b e controlled. Thus miRs are always regulated. However, some miR a really up-regulated i.e. highly expressed in specific cardiovascular diseases.
2) Line 62/63 Do all patients with LVAD produce non physiological blood-flow patterns? What isdifferentß are thes pattern still observed after device explantation?
3) Statistical method section is missing. Line 73f Where are the results of Waldtest shown in suppl. Data? I could not find them Also a description which Wald test has been used ( Single parameter, multiple parameter or non linear hypothesis) is missing in the methods section (statistics)
4) Line 102 what has dendritic spine organization and neuron projection to do with aorta
5) Lines 134f Which proteins are encoded by these genes? Increased expression of collagens might mean what? S100A4 is an intracellular protein associated with the cytoskeloton, overexpression might indeed change morphology. How does CMKLR1 fit in? Role in inflammation? Which also could be true for S100A4?
6) Lines 154ff. I think, the authors mean that some patients have endothelial dysfunction as complication. How many patients investigated had this complication? This paragraph has to be re- written . I find it difficult to understand the logic of this paragraph . last sentence Do you refer to your own findings or to literature.
7) 172ff where is the correlation analysis shown and described?
8) At the end of the discussion consequenses of findings are missing
Minor concerns
1)Line 46 probably a typing error: CV-LVAD should be CF-LVAD
2) Line 48,49 It would be nice to give an example of the „ adverse effects“
3) Lines 81 ff P values normally are designed as P, adjusted p- values should be explained, the p value which is considered as signifant should be given here or in statistical method section
4) Line 71,72 which proteins are coded by these genes. It should be stated that mRNAs encoded by these genes are meant; whereas MALAT1is a non-coding RNA.
5) For easier finding the upregulated RNAs as well GO terms should be highlighted in supplemental data . A short description of gene ontology analysis in the supplements would be nice, too . FDR should be explained
6) Description ( definition of KEGG pathway ( interaction network map?, databaseß id codeß) is completely missing , should at least be added to the supplements, so that interested readers not familiar with gene profiling could understand your work. Why data are not shown?
7) Methods The enrichment analysis tool Gorilla should be mentioned as such and and to be able to evaluate date Running modes and input should be given in supplements. The same is true for GO analysis ( reference list background frequency etc Which background st was choosen. Please define Fc, ct
8) 143ff reformulate the sentence: too long to difficult to understand.
9) all abbreviations should be defined ( written in full) when first mentioned e.e.PF-LVAD ( pulsatory flow left ventricular device) etc.
Round 2
Reviewer 3 Report
The authors responded to all my concerns properly.To my opinion the manuscript is now generally understandable and is acceptable for publication. Thank you for your explanantions. there is one minor comment: to my feelings the 5th sentence in the results section should be altered to : " Between the most twenty DEGs we identified........ to be upregulated."